# Supporting self-management for patients with Interstitial Lung Diseases: Utility and acceptability of digital devices

Malik A. Althobiani[1‡], Rebecca Shuttleworth[2‡], John Conway[3], Jonathan Dainton[3], Anna Duckworth[2,4], Ana Jorge Da Ponte[2], Jessica Mandizha[2,4], Joseph W. Lanario[2,4], Michael A. Gibbons[2,4], Sarah Lines[2], Chris J. Scotton[2,4], John R. Hurst[1], Joanna C. Porter[1], Anne-Marie Russell[2,4] *

**1** Department of Medicine, University College London, London, United Kingdom, **2** Respiratory Medicine, Royal Devon University Healthcare NHS Foundation Trust, Exeter, United Kingdom, **3** Exeter Patients in Collaboration for Pulmonary Fibrosis Research (EPIC-PF), Royal Devon University Healthcare NHS Foundation Trust, Exeter, United Kingdom, **4** Exeter Respiratory Innovations Center, University of Exeter, Exeter, United Kingdom

‡ These authors share first authorship on this work.
* a.russell4@exeter.ac.uk

**Data Availability Statement:** The fully de-identified research data supporting this publication are

## Abstract

### Introduction

Patients diagnosed with Interstitial Lung Diseases (ILD) use devices to self-monitor their health and well-being. Little is known about the range of devices, selection, frequency and terms of use and overall utility. We sought to quantify patients' usage and experiences with home digital devices, and further evaluate their perceived utility and barriers to adaptation.

### Methods

A team of expert clinicians and patient partners interested in self-management approaches designed a 48-question cross-sectional electronic survey; specifically targeted at individuals diagnosed with ILD. The survey was critically appraised by the interdisciplinary self-management group at Royal Devon University Hospitals NHS Foundation Trust during a 6-month validation process. The survey was open for participation between September 2021 and December 2022, and responses were collected anonymously. Data were analysed descriptively for quantitative aspects and through thematic analysis for qualitative input.

### Results

104 patients accessed the survey and 89/104 (86%) reported a diagnosis of lung fibrosis, including 46/89 (52%) idiopathic pulmonary fibrosis (IPF) with 57/89 (64%) of participants diagnosed >3 years and 59/89 (66%) female. 52/65(80%) were in the UK; 33/65 (51%) reported severe breathlessness medical research council MRC grade 3–4 and 32/65 (49%) disclosed co-morbid arthritis or joint problems. Of these, 18/83 (22%) used a hand- held spirometer, with only 6/17 (35%) advised on how to interpret the readings. Pulse oximetry devices were the most frequently used device by 35/71 (49%) and 20/64 (31%) measured

openly available as S1 Data and at 10.6084/m9.
figshare.24569851.

**Funding:** This work was supported by a Health
Education England Population Health fellowship to
RS. The funders had no role in study design, data
collection and analysis, decision to publish, or
preparation of this manuscript. None of the other
co-authors received specific funding for this work.

**Competing interests:** I have read the journal's
policy and the authors of this manuscript have the
following competing interests: AMR reports grants,
personal fees and other from Boerhinger
Ingelheim, personal fees from Hoffman La Roche,
outside the submitted work. AMR is a NIHR Senior
Research Leader. The views expressed in this
publication are those of the author(s) and not
necessarily those of the National Institute for
Health Research or the Department of Health and
Social Care. MAG and SL report travel grants and
honoraria from Boerhinger Ingelheim for speaking /
participation at meetings outside the submitted
work. JL has received research grants from GSK
and Astra Zeneca outside the submitted work.

their saturations more than once daily. 29/63 (46%) of respondents reported home-monitoring brought reassurance; of these, for 25/63 (40%) a feeling of control. 10/57 (18%) felt it had a negative effect, citing fluctuating readings as causing stress and 'paranoia'. The most likely help-seeking triggers were worsening breathlessness 53/65 (82%) and low oxygen saturation 43/65 (66%). Nurse specialists were the most frequent source of help 24/63 (38%). Conclusion: Patients can learn appropriate technical skills, yet perceptions of home-monitoring are variable; targeted assessment and tailored support is likely to be beneficial.

## Introduction

Interstitial Lung Diseases (ILDs) are associated with a symptom burden affecting daily life that is often complex to manage [1]. ILDs may present as multi-system disorders often alongside significant co- morbidities [2,3]. Traditionally, ILD patients have relied on self-monitoring and management methods like home oxygen therapy [4]. However, digital devices can further empower these patients by offering more granular, continuous data which can facilitate early intervention, enable detection of disease progression, and reduce healthcare costs [5,6].

Digital devices are commonly suggested for all individuals with ILD in order to improve their ability to manage and monitor their condition [7]. However, the importance of these devices is most evident for those who experience long-term breathlessness and psychological discomfort [6]. Home monitoring via commercially available devices that measure physiological parameters and symptoms may provide clinicians and patients with access to more precise continuous data on disease progression, the rate of acute exacerbations, and effects on quality of life [6–14]. This enables the development of personalized treatment approaches in this cohort [6,8–12,15]. Supported self-management measures monitor the progression of a patient's condition objectively, with digital technologies and subjectively, through the completion of patient-reported measures capturing symptom experiences and health-related quality-of-life (HRQOL) [6,12,16]. The use of digital devices, in this context, aligns with the need for early intervention and cost-effective management, reinforcing their relevance and utility for patients with ILD [6,12,15]. This approach requires support from the dedicated interdisciplinary ILD team with effective and efficient communications across the wider interdisciplinary healthcare team including primary care [7,17]. The National Institute for Health and Care Excellence (NICE) guidelines (2017), recommended that the minimally required members of an ILD team include a physician, radiologist, specialist nurse, and MDT coordinator [18]. In more complex diagnostic or treatment situations, the team should also involve thoracic surgeons, histopathologists, rheumatologists, occupational physicians, and geneticists [18].

Innovative approaches to healthcare emerged during the COVID-19 pandemic with increased interest in more readily available and affordable digital devices [7,19]. Remote monitoring programs are increasingly embedded in clinical care [7,13,14,16,20], e.g. in the UK the 'COVID Oximetry@home' service remotely monitors peripheral oxygen saturation ($SpO_2$) in patients at risk of deterioration due to 'silent' hypoxia to target care and greater efficient use of National Health Service (NHS) resources [21]. The utilization of digital health tools to assist patients with ILD, specifically, idiopathic pulmonary fibrosis (IPF), has risen in the United Kingdom between 2016 and 2022 [12,14,15,22].

It is challenging for clinicians and patients to address the healthcare needs of those living with ILD. The introduction of digital devices as an effective and efficient solution might not be the panacea some perceive. Such devices, address some needs but create others that were

unanticipated including large amounts of data that patients may require feedback on and psychosocial challenges that cannot be ignored. We know that these patients use devices to self-monitor their health and well-being but, little is known about the range of devices, selection, frequency of use and overall utility [6,14,23]. We aimed to characterize the types of digital devices used by people with ILD for home-monitoring purposes, outside remote monitoring programs. This work was subjected to critical peer review by the Respiratory Specialty Governance Group and registered as a service improvement project at Royal Devon University Hospitals NHS Foundation Trust/UK (ID:20–4946).

## Methods

We developed a cross-sectional survey with our patient partners (JC and JD) in the Exeter Patients in Collaboration for Pulmonary Fibrosis Research (EPIC-PF) group. The content and phrasing of the questions in the survey build evolved iteratively through discussions with ILD expert clinicians and patient partners via a series of online meetings over 6-months. The interdisciplinary supported self- management group at Royal Devon University Hospitals NHS Foundation Trust critically appraised the penultimate survey, built using the NHS-affiliated Survey-Monkey.com. The 48-question survey was divided into 5 sections: 1. respondents' characteristics / demographics, 2. types of devices in use, 3. frequency of device use, 4. technology used and 5. factors influencing choice and use of devices. If a respondent answered that they did not use a certain type of device, skip logic was applied eliminating the subsequent more detailed questions relating to this specific device. A further question requested additional comments to capture qualitative experiential data.

A survey preamble of clear concise instructions provided email contact details for the survey team for any concerns or queries. The brief clearly stated that the survey was specifically for people diagnosed with ILD; those who were not diagnosed with ILD were consequently excluded from the study. This international survey was open for participation from September 2021 to December 2021. It was anonymous with clear information on how data collected would be used. Respondents had the option to add their contact details at the end of the survey to receive an update on the results. Assistance with translation and accessibility of the survey was available via email contact.

The direction of response categories was consistent for ease and included a mixture of binary and 4- point Likert scaled responses. Four response categories are regarded as the absolute minimum and selecting an even number removed the predilection for the middle ground [24]. The survey started with general questions to qualify respondents and introduce the topic followed by specific questions and finishing with general, easy-to-answer demographic questions. This approach was to promote engagement, decrease drop-out rates and improve the quality of the experience. We piloted the survey in our EPIC group to assess structure and flow prior to distributing via national ILD support groups and the social media channel (Twitter 'X'). The survey was open for three months and re-promoted at monthly intervals.

We present data by the number of respondents and percentage proportionate to the total number of responses per item. Qualitative data, transcribed according to accepted practice underwent thematic analysis, coding and organization using NVIVO (Lumivero formerly QSR International) software [25]. The study team consisted of three mixed methods researchers experienced in qualitative approaches and survey design (AMR; MA and RS). Informed consent from participants was obtained prior to opening the questionnaire.

## Results

### Respondents' characteristics/demographics

Respondents from Europe, Asia, USA and Canada (n = 104) participated with 15 respondents excluded, as they did not have a diagnosis of ILD (Table 1). Eighty-nine respondents completed the survey. Forty-six (52%) were diagnosed with IPF. Thirty-four (38%) reported having a diagnosis for more than 5 years whilst 39% held a diagnosis 2–5 years (SI-1).

Breathlessness was a dominant symptom graded on the MRC breathlessness scale; moderate (grade2-3) for 29/65(45%), and severe (grade 4–5) for 33/65(51%). Sixty (67%) reported using medication for ILD. Sixty-five (73%) reported comorbidities to include arthritis or joint problems 32(49%), followed by high blood pressure 20(31%), and heart disease 13(20%).

### Contact with clinicians

A majority of respondents (70/89(79%)) reported having contact every 3–6 months with a specialist ILD doctor or nurse. For 18(20%) contact was yearly, the remaining 1% >18months. Several participants expressed frustration over this "Have been left on my own" "Feeling abandoned" "Unfortunately I have not had contact with respiratory department since December 2020, and that was by telephone". Some participants hoped for ". . .monthly contact just to discuss condition. . . via Zoom or phone [if only] for guidance". Sixty-five respondents reported contacting their General Practitioner (GP) or ILD team for the following reasons: increased breathlessness (66%) low saturation on home monitor with worsening symptoms, (49%) change in sputum, (48%) increased cough and (32%) concerns about medication side effects.

**Table 1. Respondents' characteristics.**

| Gender | % | N |
|---|---|---|
| Male | 33.7% | 30 |
| Female | 66.3% | 59 |
| Continent | | |
| Europe—UK | 80.0% | 52 |
| USA | 1.5% | 1 |
| Canada | 4.6% | 3 |
| Asia | 13.8% | 9 |
| UK Region | | |
| London and home-counties | 11.5% | 6 |
| Southeast | 13.5% | 7 |
| Southwest | 26.9% | 14 |
| Midlands | 26.9% | 14 |
| Northwest | 3.8% | 2 |
| Scotland and Highlands | 11.5% | 6 |
| Northern Ireland | 1.9% | 1 |
| Other | 3.8% | 2 |
| Age | | |
| 18 to 40 | 6.7% | 6 |
| 41–50 | 10.1% | 9 |
| 51–60 | 19.1% | 17 |
| 61–70 | 29.2% | 26 |
| 71–80 | 23.6% | 21 |
| 80–85 | 10.1% | 9 |
| >85 | 1.1% | 1 |

## Devices

**Finger pulse oximeter.** Seventy-six (86%) participants used devices to monitor their vital signs, mainly a combination of finger probe peripheral oxygen saturation and heart rate 43 (61%) followed by blood pressure 37(52%) (Table 2). Thirty-four (47%) reported measuring oxygen saturation (SpO2) when their symptoms 'felt' bad, followed by 30(42%) who regularly measured SpO2 regardless of how they felt. Only 4(6%) reported wearing a device throughout the day that was constantly measuring saturation. Forty-three (61%) noted low SpO2 during exertion whilst 40(46%) recorded low SpO2 readings when feeling breathless or unwell. For 17 (24%) SpO2 readings remained within the normal range despite feeling breathless or unwell (Table 1). 22(31%) sought information about their device online or watched an online video. Eighteen (25%) felt they did not need any advice regarding the use of home oximetry. The minority of respondents received advice from either their GP or ILD nurse specialist.

**Home-spirometer.** A minority, 18/83 (22%) reported using home-based spirometry devices, contrary to our expectations. Of the 18(22%) using home-spirometry, 14(88%) measured their forced vital capacity (FVC) with most respondents (n = 13) not experiencing any difficulty in getting reproducible readings, while 3 participants noted that the spirometer was "hard work and difficult to do" with one reporting: "When I exhale, I find it difficult not to cough, which can affect the readings significantly and it is quite exhausting when repeating the process." "It's hard work and difficult to do it the same each time". For those using home-spirometry 11 respondents were either very or somewhat confident of their technique whilst 5 felt very unconfident and 2 abstained. Six respondents reported they had received advice predominantly from their healthcare professional (n = 5). Twelve respondents used home-spirometry in conjunction with changes in their symptoms to prompt communication with their clinical team, whilst 4 only did so following 2–3 successive declines in spirometry readings.

**Technologies:** All participants had internet access at their own home 89(100%) with 84 (95%) using a smartphone, 71(81%) a tablet computer and 63(72%) a laptop regularly. Twenty-eight (35%) reported using smart / fitness watch and activity trackers, mostly Apple, 10(43%). Twenty-nine (28%) reported using smart phone/tablets and apps for monitoring their health. The most frequently reported health monitoring apps were Apple health 4(22%), Spirobank 4(22%), ViHealth 4(22%) and Fitbit 3(16%). Whilst we recommend home-monitoring devices are CE marked only 34(44%) were able to confirm this, suggesting this may not be a priority for users.

**Frequency of use of each device:** Forty-six (72%) respondents reported taking home-monitoring measurements while sitting or resting and 21(33%) walking inside the home (Fig 1).

**Financial support and motivation.** Only 9(12%) reported that their GP or specialist team supplied their devices. 22(32%) purchased their own informed by online reviews, followed by 12(17%) informed by ILD forum/support groups and 10(14%) on the recommendation of a healthcare professional. An additional 6 (9%) chose their devices because it was the lowest price they could find. The motivation for using health apps and devices had a strong psychological component associated with control 25(40%) as well as a marker of progression 19(30%) and validating 'feeling unwell' 25(40%); (Table 3).

**Thematic analysis.** Three overarching themes emerged identified as 'facilitators', 'barriers','experiences' and 'impacts' of using home-monitoring applications. Within the overarching themes further subthemes were identified.

**Overarching Theme 1: Facilitators to using home monitoring in patients with ILD.** Fifty-two respondents reported positive effects of using home-monitoring devices, gaining "reassurance" "affirmation of symptoms" "peace of mind" "control" and "self-management support" during home-monitoring (Fig 2).

**Table 2. Home devices in use.**

| Study Variables and Responses | % | N |
|---|---|---|
| **Uses devices to monitor vital signs** | 86% | 76 |
| **Does not use devices to monitor vital signs** | 14% | 12 |
| **Type of devices used to monitor vital signs** | | |
| Finger probe oxygen saturation monitor (singularly) | 49% | 35 |
| Heart rate monitor (singularly) | 10% | 7 |
| Combined finger probe saturation and heart rate monitor | 61% | 43 |
| Blood pressure monitor | 52% | 37 |
| **Advice on how to use devices that monitor vital signs** | | |
| I was given advice by my GP | 11% | 8 |
| I was given advice by my local support group | 7% | 5 |
| The pharmacy explained to me | 3% | 2 |
| I read information online and/or watched an online video | 31% | 22 |
| I was given advice by the ILD nurse specialist | 13% | 9 |
| I have had no advice/education and would welcome some | 13% | 9 |
| I have not felt the need for any advice/education | 25% | 18 |
| **Frequency of SpO2 monitor** | | |
| I regularly measure my oxygen saturations regardless of how I feel | 42% | 30 |
| I measure my oxygen saturations if my symptoms are bad | 47% | 34 |
| I measure my saturations when exerting myself (e.g. going up stairs/ exercising) | 29% | 21 |
| I measure my readings after exertion whilst I am recovering | 29% | 21 |
| I measure my saturations when I am at rest | 21% | 15 |
| I wear a device throughout the day that is constantly measuring my saturations | 6% | 4 |
| **SpO2 results match symptoms** | | |
| When I am feeling more breathless or unwell my saturations are often low | 56% | 40 |
| My saturations are low, but my symptoms are stable | 7% | 5 |
| My saturations are low when I am exerting myself | 61% | 43 |
| When I feel breathless and/or unwell my saturations can be within normal range. | 24% | 17 |
| **Spirometer** *(Aspect of lung function monitored)* | | |
| Forced vital capacity (FVC) | 88% | 14 |
| Forced expiratory volume in the first second (FEV1) | 63% | 10 |
| Flow volume loop | 44% | 7 |
| FEV1 /FVC ratio | 44% | 7 |
| **Confidence of spirometer technique** | | |
| Very unconfident | 25% | 4 |
| Somewhat unconfident | 6% | 1 |
| Somewhat confident | 38% | 6 |
| Very confident | 31% | 5 |
| **Advice on interpreting spirometry readings** | | |
| **Advice received** | 35% | 6 |
| **No advice received** | 65% | 11 |
| I'm unsure | 35% | 27 |
| **Position when you take measurements with home-monitoring devices** | | |
| Sitting or resting | 72% | 46 |
| Walking inside | 33% | 21 |
| Walking outside | 28% | 18 |
| Cycling | 9% | 6 |
| Running | 0% | 0 |

*(Continued)*

**Table 2.** (Continued)

| Study Variables and Responses | % | N |
|---|---|---|
| Swimming | 0% | 0 |
| Exercise / rowing machine | 8% | 5 |
| Gardening | 6% | 4 |

## Subthemes

**Reassurance:** Eighteen respondents felt reassured by their use of home-monitoring devices, particularly in relation to oxygen saturation monitoring when oxygen levels return to normal levels: "I feel I can take the panic away with an oxygen monitor because I can sit and breathe and see my oxygen level rise" (ID 18, Female). Another participant felt reassured "[I] can keep an eye on oxygen levels to reassure myself or know when to get help" (ID 85, Female).

**Symptom affirmation:** Fifteen respondents commented on how they felt a sense of affirmation by their use of home-monitoring devices. The biggest expressed benefit was in knowing when to seek medical help or attention from self-monitored results: "helps you know when you need intervention" (ID 45, Male) and "You know if there is an issue" (ID 52, Female).

**Control:** Seven respondents experienced a sense of control in using home-monitoring devices described as the reduction of anxiety around the illness. "Helps me stay calm when I can see it rising after exertion" (ID 25, Male). Being able to monitor symptoms at home on a regular basis made some participants feel in control and less anxious after episodes of severe breathlessness "Makes me feel I control of this illness" (ID 27, Female) and "Stops me panicking and wasting GP time" (ID 70, Female).

**Self-management support:** Fourteen respondents described self-management support, as empowering. They reported positive experiences in being able to track and trend their progression over time, "I am able to monitor my situation, helps to keep track" (ID 64, Female).

**Overarching Theme 2: Barriers to using home monitoring in patients with ILD.** Thirty-four participants reported negative effects with home-monitoring devices bringing "worries"

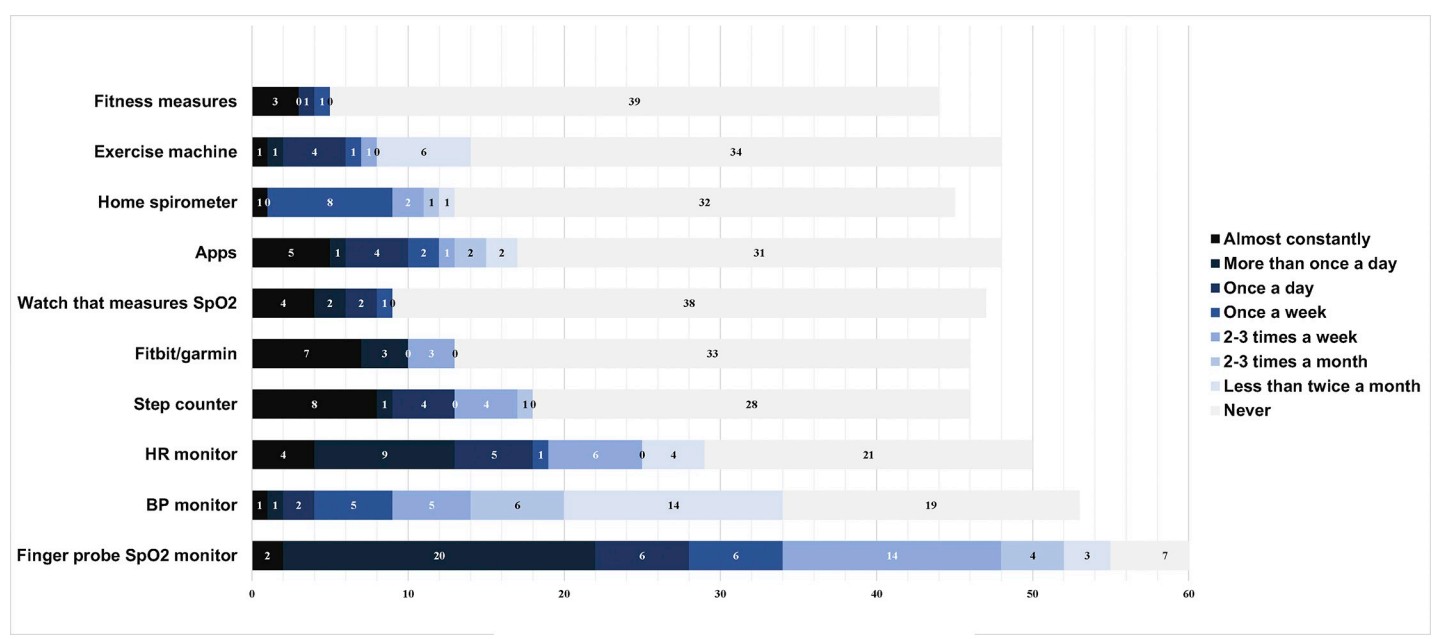

**Fig 1. Frequency of using home monitoring devices.**

**Table 3. Reasons for using apps and home-monitoring devices.**

| Reason for using apps and home-monitoring devices. | % | N |
|---|---|---|
| My Doctor/Clinical Nurse Specialist/Other healthcare professional recommended it | 14% | 9 |
| It makes me feel more in control of my illness | 40% | 25 |
| It helps me to quantify when I feel unwell | 40% | 25 |
| It helps me monitor the progression of my disease | 30% | 19 |
| I feel it picks up changes earlier that can be acted on | 22% | 14 |
| It gives me reassurance | 46% | 29 |
| It helps me know when to call for help | 24% | 15 |
| It helps me know when to use my home oxygen | 24% | 15 |
| Not applicable—I do not use any apps or home-monitoring devices | 22% | 14 |

The additional comments collected in the survey gave further insights regarding the respondents' experiences using home-monitoring devices. These data underwent thematic analysis and are discussed below within a framework of enablers and barriers to home spirometry.

"anxiety" in part due to "inaccurate" readings and "insignificant support" i.e. self- management without a 'supportive' component and / or technical challenges with devices (Fig 3).

## Subthemes

**Anxiety and worry:** Respondents reported barriers and concerns using home-monitoring devices finding them to be disturbing, worrisome, inaccurate, burdensome, and complicated. One reason people cited for not using home-monitoring is that it negatively affected the way they thought of their illness.

Some negative effects of self-monitoring include developing an obsession with readings or living with a constant reminder of the illness. A sense of worry for some participants was that readings would be negative. Participants' explanations ranged from anxiety and worries "Tend to unnecessarily get stressed on fluctuating readings, it makes you paranoid" (ID 48 Male); another participant said, "Makes you worry if your vitals are not where they should be" (ID 45 Male) and a third said, "Readings can be low without increased breathlessness leading to raised anxiety" (ID 86 Female). Some respondents reported lack of confidence in the accuracy of information provided by home-monitoring devices. "Measurements of oxygen saturation are difficult it seems 2 points lower than the one I use at hospital" (ID 32 Male). Another commented on feeling 'ill equipped' to monitor himself well.

**Supportive Self-Management:** Some respondents felt insufficient support was provided by clinical teams. Three key themes emerged; barriers to access (for some devices are too expensive n = 12) feedback on data and inadequate knowledge/ information with some respondents lacking confidence in interpreting the readings (n = 8) (Table 3). One participant overcame these barriers "I am a scientist, so I know value in data and went ahead to manage myself" **(ID 31 Female).** [Table 4]

**Technical challenges with devices:** Respondents identified some challenges with home-monitoring devices, from a lack of familiarity and knowledge of digital device usage to lack of training. "I find the technology is a little tricky" (ID 96 Male) "Spirometry is hard work and am not confident I do it well" (ID 101 Male) "Not sure how to interpret readings" (ID 37 Male). "Won't work and am not confident I do it well" (ID 101 Male). Another participant stated, "[I] use a plotted pressure monitor, but have no real idea what my BP should be or what the figures actually mean" (ID 78 Female). To summarize, respondents felt as if they needed more education or an explanation of what they were trying to look at and achieve.

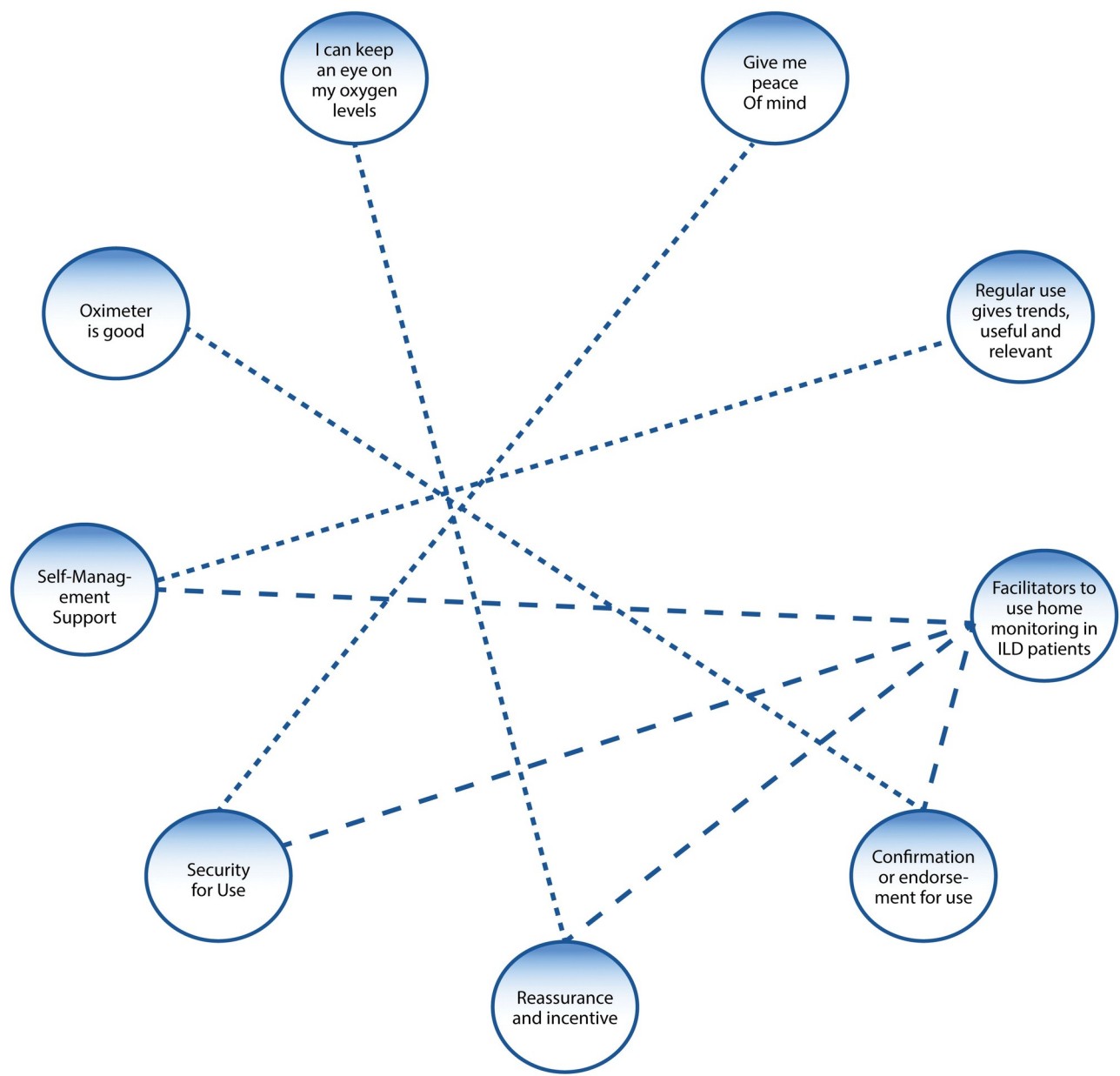

**Fig 2. Facilitators to use home-monitoring in patients with ILD.**

Further, results indicate that participants experienced difficulties in understanding and interpreting the results of health monitoring captured in the following quotes: "hope I am reading it okay" (ID 37 Female), "I don't have a clue what they mean" (ID 78 Female), "not sure what to do when sats in low 80's'" (ID 98 Female), "Spirometry is hard work and I am not confident, I have no idea if this is the right thing to do" (ID 78 Female), "I have no idea what my BP should be and what the figures actually mean" (ID 78 Female), and "not sure how to interpret readings" (ID 11 Female).

The utilization of multiple digital devices for health monitoring presents a challenge for patients with ILD, as has been reported in the following quotes: "calluses on fingers. Doesn't work if readings are low and different to hospital readings" (ID 12 Male). "Not sure of

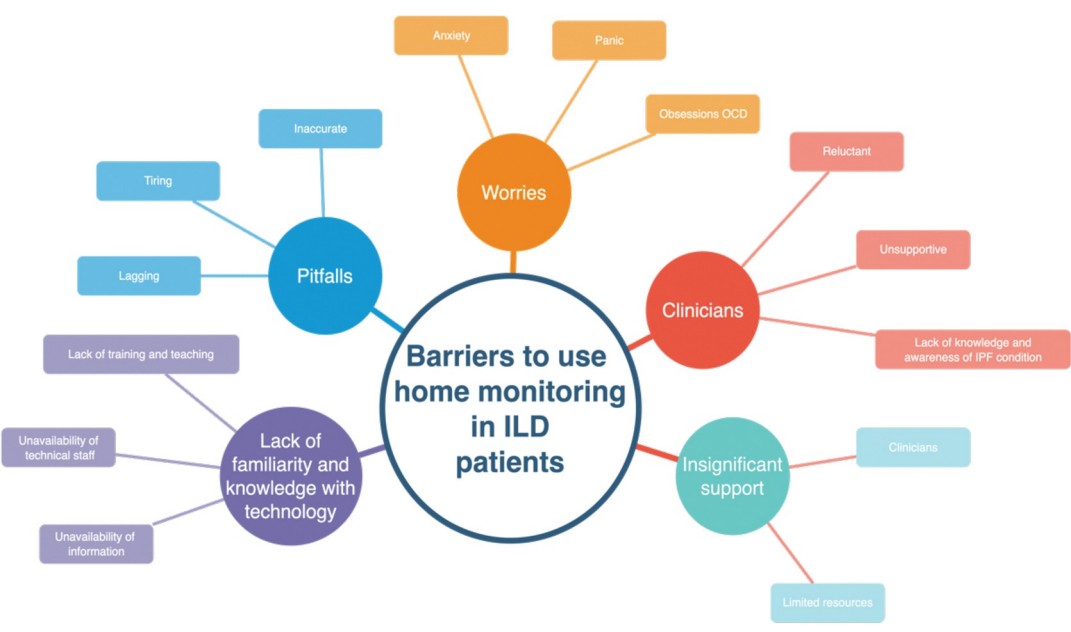

**Fig 3. Barriers to using home-monitoring in ILD patients.**

accuracy" (ID 21 Male). "Spirometer isn't accurate". (ID 81 Male) "Measurements of O2 sats are difficult" (ID 32 Male). "HR doesn't work for me" (ID 73 Male). "Device doesn't detect DLCO" (ID 88 Male). "Too complicated to set up" (ID 13 Female). "I find technology tricky" (ID 96 Male).

**Overarching Themes 3&4: Experiences of using applications and the impact on condition management.** Forty of the 63 respondents reported home-monitoring to be somewhat helpful; and 18 very helpful in managing their condition particularly mobile applications were "useful" "enabling help-seeking behaviors" "record Management" and "exercise motivation" (Fig 4).

## Subthemes

**Usefulness/enabling help-seeking behaviors:** Respondents found applications in self-monitoring to be useful tools to monitor "vital signs and physical activity" (ID 84 Female). "The

**Table 4. Barriers to use of home-monitoring in people living with ILD.**

| Access | Feedback | knowledge / information |
|---|---|---|
| "I want to feel well supported with easy and quick access to specialist ILD professionals that I can speak to if my symptoms or home-monitoring results change. Unless this is available, anxiety can occur more frequently than without home-monitoring" (ID 88 Male) "would appreciate monthly contact to discuss condition" "no one to talk to"(ID 81 Male). | "Would be good if they were reviewed periodically by a specialist' "I send my spirometry results off and hear nothing as if they have disappeared into a black hole. Some feedback would be welcome." (ID 87 Male). 'I don't contact anyone as they're not interested'(ID 6 Female) 'problems I pick up are ignored' 'what's the point in telling them' (ID 18 Female) 'In the past medical staff question my results against their readings so I stopped reporting back"(ID 92 Female). | 'I feel very let down around the information side of things. I feel there is plenty of information and support for COPD but very little support in my case for ILD" (ID 85 Female) 'lack of knowledge about PF by GP's and receptionists' (ID 92 Male) 'HCW reluctant for me to monitor at home' 'mocked' (ID 32 Female). |

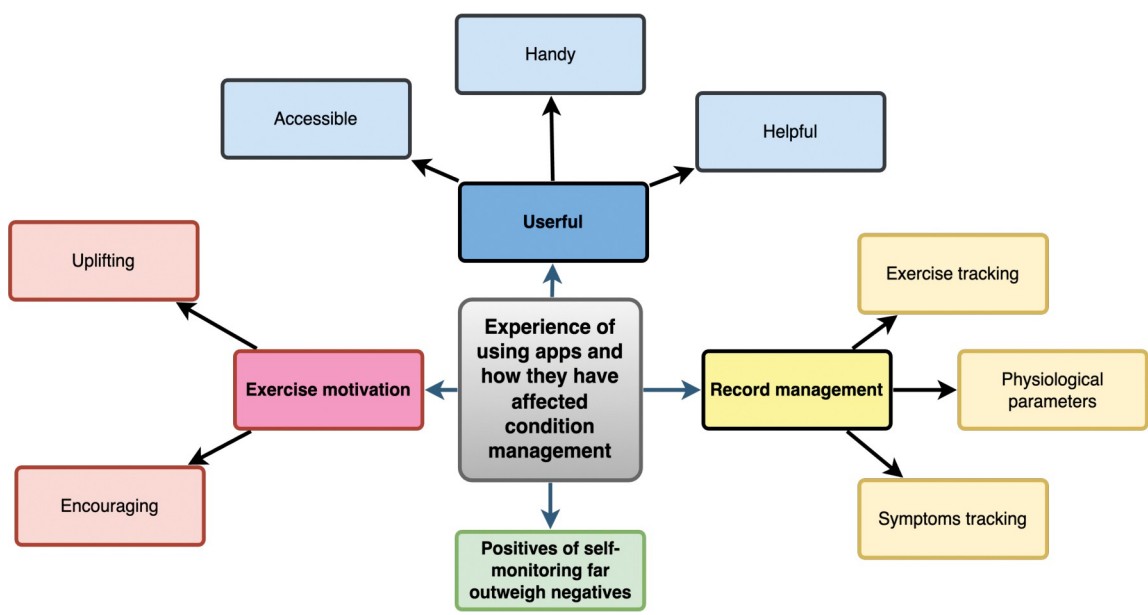

**Fig 4. Experiences of using applications and the impact on condition management.**

applications I use, also empower me to talk about my disease with HCP and changes I have noted" (ID 104 Male). "My Kardia application is usually reassuring when I think I have gone into Atrial fibrillation", occasionally I have gone to A&E and taking the printout seems to help the staff" (ID 96 Male). "Handy for monitoring vital signs and helps monitor my walking" (ID 35 Male). Respondents found devices and home-monitoring were a tool to assist help-seeking: "[they] let me know if I need medical attention" (ID 12 Male); to pick up changes: "detect an exacerbation before it happens" (ID 104 Male) and to support communication with healthcare professionals: "Good to share with 111" (ID 31 Female); "low sats got me seen in hospital in 2 days" (ID 92 Male); "empowerment to talk about my disease with HCP and changes I have noted" (ID 104 Male). (Fig 4).

**Record Management:** Applications help to keep track of results and make access to records easy for participants and clinicians. One person liked: "to have a record of past results for comparison with more recent results" (ID 84 Female). Through record keeping people reported being able to "baseline" (ID 42 Male) and "keep track" (ID 64 & 31 Females) to "identify changes (ID 88 Male) and "take action" (ID 79 Male). Respondents felt positive outcomes of self-monitoring and record keeping in that they were enabled to quantify symptoms, which in turn helped with both anxiety management and help-seeking behavior.

**Exercise Motivation:** Respondents reported that using applications encourages physical activity: "Encourages me to go the extra few hundred yards", "Good to keep an eye on how active I am" (ID 101 Male), "enables me to exercise safely (ID 104 Male)".

## Discussion

This survey adds to existing literature in ILD on the perspectives and experiences of home-monitoring for healthcare providers, including the prevalence of use, experience, contemporary methods, enablers, and barriers [13,20,26,27]. A major contribution of this survey is to extend the research to cover people living with ILD, including patients' usage and experience of a range of devices, and the utility of digital devices. Although the results of this mixed-methods survey have demonstrated that digital devices are widely used among patients with ILD,

the views and perspectives regarding the use of these devices is varied. The introduction of digital devices as an effective and efficient solution might not be the panacea some perceive. Digital devices address some needs but create others that were unanticipated, including large amounts of data that patients may require feedback on and psychosocial challenges that cannot be ignored.

Our findings show that interest in digital devices seems to stem from the perceived positive effects of using home-monitoring devices, which have provided "reassurance", "confirmation", "peace of mind" and "self-management support" during home-monitoring. In contrast, some respondents have also found home-monitoring devices to bring "worries" and "anxiety", in part due to "inaccurate" and "insignificant support". Specifically, patients with IPF where the condition had progressed to a certain stage where symptoms were more apparent and where carrying out the task (e.g. spirometry) would trigger coughing and give unreliable results, which demotivated and discouraged further use.

Our findings mirror those of previous studies, in which remote monitoring was reported to be acceptable among patients with ILD [7–9,11,12,14,28–30]. In a 24-week randomized control trial, Moor et al. [11] demonstrated that home-monitoring was appreciated and highly acceptable to patients. Edwards et al. [8]. used a mobile app and home spirometry in patients with IPF for six weeks and found that patients were happy and wished to continue using the apps at the end of the study.

The findings further elaborate on how self-monitoring using digital devices helps patients with ILD to contact their GP or ILD specialists regarding medication side effects and therapy response. Broos et al. [31] and Moor et al. [11] similarly demonstrated that home-monitoring assisted continuous evaluation of the patient's response to therapy and allowed for individualized treatment adjustment.

Our results also illustrate that digital devices allow patients with ILD to exercise control over their health. The greatest benefit expressed was knowing when to seek medical help or attention due to self-monitored results. Our findings agree with previously published studies, demonstrating that involving patients in monitoring their own health gives them feeling of being in control over their own health status [9,32,33]. Respondents were reassured by their use of home-monitoring devices, particularly in relation to oxygen saturation monitoring. This implies that digital devices may have an effect on the daily experiences of patients with ILD. This valuable finding sheds light on the need to continuously support future users of digital devices.

Participants reported that utilizing mobile applications and wearables can be an effective way to promote physical activity. Wallaert et al. [34] and Root et al. [35] highlighted the importance of daily physical activity for patients with IPF in managing their condition, whilst Bahmer et al. [35,36] demonstrated that there is a link between progression of IPF and decreased physical activity. These digital devices can be useful in providing patients with real-time feedback on their physical activity levels, personalized goals, and reminders to stay active, which can improve adherence to exercise regimes and enhance the quality of life for patients with IPF.

Despite the encouraging results of using digital devices, our study found several challenges and obstacles in relation to the use of home-monitoring devices among patients with ILD and IPF, more specifically. These challenges and obstacles experienced by patients are consistent with prior studies of patients with ILD [14,37]. Our findings revealed that the use of home-monitoring devices may create additional anxiety for patients with ILD. Many patients reported feeling anxious about low readings from the devices, even when they were not experiencing increased breathlessness. It is worth noting that anxiety is a common comorbidity among people with IPF [38]. Additionally, patients have expressed doubts about the

accuracy of the information provided by home-monitoring devices, and this lack of confidence can be further intensified by a lack of support. Consistent with previous research, the results reveal that digital devices may influence self-perception and self-image in relation to health status. Self-monitoring via home-based digital devices is related to an obsession with readings and is found to be a constant reminder of living with an illness and diagnosis [23,37]. Thus, it can be a source of anxiety and worry among patients with ILD [39,40].

Secondly, our findings show that a lack of adequate support, such as insufficient training, barriers to access, and inadequate knowledge among non-specialist staff, can impede the effectiveness of remote monitoring interventions. Moor et al. [11] found that providing sufficient training and easy access to a helpdesk for technical issues, as well as real-time alerts and feedback, significantly increased adherence to home-monitoring with handheld spirometry during a 24-week randomized controlled trial [11]. As a result, we argue that it is essential for remote monitoring programs to include a dedicated hotline and helpdesk for technical support and that there is a need for support groups for ongoing assistance. However, alternative forms of support, such as email or asynchronous communication, should also be considered to ensure that the program is cost-effective, while still providing the necessary support to participants.

Thirdly, our results identified technical difficulties as a major challenge for patients with ILD, specifically in terms of the lack of technical support to address the troubleshoot these issues. While previous studies have shown that both daily [8,10,12,29,41], and weekly [7,9] home based spirometry measurements are acceptable and easy [42], our findings show that these technical difficulties can impede the use of these devices. These findings are consistent with previous research on remote monitoring of ILD patients, where Maher and colleagues [41,43] reported technical problems that prevented the primary endpoint analysis of spirometry measurements.

## Future implications

Patients with ILD are at risk of disease progression including deterioration of lung function over time [44]. The risk of disease progression depends on the specific type of ILD, as well as the individual patient's disease course [45]. Early access to digital devices during the treatment pathway, while individuals are actively seeking information and education, has the potential to provide greater benefits and encourage positive lifestyle changes [16]. The use of digital devices is becoming integral to the overall monitoring and support framework and standard of care for patients. Such approaches can empower patients to measure and monitor the progression of their condition objectively. Using commercially available devices (e.g. wearables, handheld spirometry, non-contact monitors, and mobile apps) could allow early intervention for managing ILD and slowing the rate of deterioration.

We must be mindful of difficulties in using devices or getting reliable results for individuals where the condition had progressed or when symptoms are more apparent. Of the 58 respondents who added comments 29 reported that the measurements they obtained were variable, 18 that these were stable. To ensure the success of these interventions, our findings highlight the importance of addressing the following:

- Adequate financial resources

- Meaningful patient involvement in the design of supported self-management programs

- Shared decision-making to inform adaptation/cessation of device usage when the burden may outweigh the benefits.

- Accessibility of cost-effective home-monitoring equipment

- Training in the utilization of digital devices and interpretation of data for patients and clinicians

- Technical support for addressing issues related to digital devices

- Effective communication between healthcare providers and patients regarding results and timely action planning

- Need for more evidence e.g. RCTs

## Strengths and limitations

The strength of this survey is that it captures both qualitative and quantitative information on remote monitoring in patients with ILD and is the first of its kind in ILD in the UK. It demonstrates an appetite and motivation to self-monitor. The survey has some limitations, for example, the representativeness of the sample, as the participants were predominantly from the UK and self-selecting. The distribution was exclusively electronic and reliant on non-governmental organizations and Twitter so necessitated some social engagement and digital literacy. Whilst men use the Internet more for informational purposes, women's usage is reported to be more for social and expressive purposes [46], which may explain the higher female response rate in this survey, unexpected in an ILD population, particularly IPF a male dominant condition. Whilst people's use of social media, and other methods has increased as a consequence of the COVID-19 pandemic, we may not have captured the perspectives of lower-income and minority populations. Notably, we did not receive any request for translation of the survey.

## Conclusion

We aimed to quantify patients' usage and experiences of electronic devices in ILD. Our survey and analysis illustrate the extent to which patient expertise has the potential to drive health system priorities. Having lived through the COVID-19 pandemic, people with complex health conditions have learned how to navigate the challenges of self-monitoring and are influencing how we deliver routine healthcare. Our work demonstrates that digital approaches, are an important component of a supported self-management program but further work is needed to develop bespoke digital pathways of care tailored to individual needs. This requires solution focused thinking to meet the needs of wide ranging digital abilities in those with chronic health conditions and political commitment to optimize wifii connections for those in rural and harder to reach communities, the very people most likely to benefit from supportive self-management approaches. Health economic evaluation of digital care pathways is very much needed alongside an in-depth exploration of the impact of home monitoring using digital devices on the family dynamic. Digital devices will continue to be embedded in our lives and they yield clinically useful data as a component of not a replacement for traditional models of healthcare.

## Supporting information

**S1 Data. Digital devices ILD survey de-identified data set.** CSV files contains raw survey data questions as asked and participants responses available at 10.6084/m9.figshare.24569851 (CSV)

## Acknowledgments

We are grateful to the Exeter Patients in Collaboration for Pulmonary Fibrosis Research (EPIC-PF) group for their support with the survey design and distribution.

## Author Contributions

**Conceptualization:** Malik A. Althobiani, Rebecca Shuttleworth, Anne-Marie Russell.

**Data curation:** Malik A. Althobiani, Rebecca Shuttleworth, Jonathan Dainton, Ana Jorge Da Ponte, Jessica Mandizha, Joseph W. Lanario, Sarah Lines, Anne-Marie Russell.

**Formal analysis:** Malik A. Althobiani, Rebecca Shuttleworth, John Conway, Jonathan Dainton, Anne-Marie Russell.

**Funding acquisition:** Rebecca Shuttleworth, Anne-Marie Russell.

**Investigation:** Rebecca Shuttleworth, Jonathan Dainton, Anna Duckworth, Jessica Mandizha, Anne-Marie Russell.

**Methodology:** John Conway, Jonathan Dainton, Anna Duckworth, Anne-Marie Russell.

**Project administration:** Rebecca Shuttleworth, John Conway, Anna Duckworth.

**Resources:** Anne-Marie Russell.

**Software:** Rebecca Shuttleworth.

**Supervision:** John R. Hurst, Joanna C. Porter, Anne-Marie Russell.

**Validation:** Malik A. Althobiani, John Conway, Michael A. Gibbons, Chris J. Scotton, John R. Hurst, Joanna C. Porter.

**Visualization:** John Conway, John R. Hurst.

**Writing – original draft:** Malik A. Althobiani, Rebecca Shuttleworth, Jonathan Dainton, Anne-Marie Russell.

**Writing – review & editing:** Malik A. Althobiani, Rebecca Shuttleworth, John Conway, Anna Duckworth, Ana Jorge Da Ponte, Jessica Mandizha, Joseph W. Lanario, Michael A. Gibbons, Sarah Lines, Chris J. Scotton, John R. Hurst, Joanna C. Porter, Anne-Marie Russell.

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
