## [Decision Letter · Decision Letter 0]

31 Aug 2023

PDIG-D-23-00257

Towards self-management for patients with Interstitial Lung Diseases: Utility and Acceptability of Digital Devices

PLOS Digital Health

Dear Dr. Russell,

Thank you for submitting your manuscript to PLOS Digital Health. After careful consideration, we feel that it has merit but does not fully meet PLOS Digital Health's publication criteria as it currently stands. Therefore, we invite you to submit a revised version of the manuscript that addresses the points raised during the review process.

Please submit your revised manuscript within 60 days Oct 30 2023 11:59PM. If you will need more time than this to complete your revisions, please reply to this message or contact the journal office at digitalhealth@plos.org. Please include the following items when submitting your revised manuscript:

We look forward to receiving your revised manuscript.

Kind regards,

Haleh Ayatollahi

Section Editor

PLOS Digital Health

Journal Requirements:

1. Please send a completed 'Competing Interests' statement, including any COIs declared by your co-authors. If you have no competing interests to declare, please state "The authors have declared that no competing interests exist". Otherwise please declare all competing interests beginning with twhe statement "I have read the journal's policy and the authors of this manuscript have the following competing interests:"

2. In the Funding Information you indicated that no funding was received. Please revise the Funding Information field to reflect funding received.

3. Please send a completed 'Competing Interests' statement, including any COIs declared by your co-authors. If you have no competing interests to declare, please state "The authors have declared that no competing interests exist". Otherwise please declare all competing interests beginning with twhe statement "I have read the journal's policy and the authors of this manuscript have the following competing interests:"

4. Please amend your detailed Financial Disclosure statement. This is published with the article. It must therefore be completed in full sentences and contain the exact wording you wish to be published.

5. Please provide separate figure files in .tif or .eps format only and remove any figures embedded in your manuscript file. Please also ensure that all files are under our size limit of 10MB.

6. We ask that a manuscript source file is provided at Revision. Please upload your manuscript file as a .doc, .docx, .rtf or .tex.

7. In the online submission form, you indicated that "Analysis was done in Excel and NVIVO - Raw anonymised data is available on direct request to the corresponding author for legitimate non-commercial purposes". All PLOS journals now require all data underlying the findings described in their manuscript to be freely available to other researchers, either 1. In a public repository, 2. Within the manuscript itself, or 3. Uploaded as supplementary information.

Additional Editor Comments (if provided):

The manuscript was interesting. Please consider the following comments in your revision.

1- Please follow the journal style for formatting the abstract and the manuscript.

2- In the methods section, please add adequate information about the study participants and sampling methodology.

Reviewers' comments:

Reviewer's Responses to Questions

**Comments to the Author**

1. Does this manuscript meet PLOS Digital Health’s publication criteria? Is the manuscript technically sound, and do the data support the conclusions? The manuscript must describe methodologically and ethically rigorous research with conclusions that are appropriately drawn based on the data presented.

Reviewer #1: Partly

Reviewer #2: Yes

Reviewer #3: Partly

2. Has the statistical analysis been performed appropriately and rigorously?

Reviewer #1: Yes

Reviewer #2: N/A

Reviewer #3: Yes

3. Have the authors made all data underlying the findings in their manuscript fully available (please refer to the Data Availability Statement at the start of the manuscript PDF file)?

Reviewer #1: Yes

Reviewer #2: Yes

Reviewer #3: No

4. Is the manuscript presented in an intelligible fashion and written in standard English?

Reviewer #1: Yes

Reviewer #2: Yes

Reviewer #3: Yes

5. Review Comments to the Author

Reviewer #1: General

1. There are capitalization errors, punctuation errors, missing symbols (trademark or copyright, such as for commercial software and technology) and run-on sentences throughout (not further identified in this review)

2. While the purpose is identified, the abstract lacks sufficient details regarding clear statement of study objectives, I/E criteria for respondents, survey tool validation, timeline, data areas/elements, content areas, and data analysis (descriptive, hypothesis testing, etc). It is unclear if the responses were anonymous. Without these elements, the data summary appears to be somewhat random and it is unclear if the conclusions address the objectives and/or are justified. 

3. Abstract (and elsewhere) states 65 respondents completed the survey, but reports data on the 89 that met eligibility criteria. It is therefore unclear if survey completion is needed for data inclusion. The denominator for responses is inconsistent, implying incomplete survey or data subsets being presented. Because of the changing denominators, they should be consistently reported throughout the results section as well.

4. Much of the data summarized in the results narrative is available in summary form elsewhere and is difficult to follow. Demographics could best be summarized by a table.

5. Given the variable denominators for responses, it is essential to indicate the total number of responses within a category so that the percentages can be verified. In come cases, data within a topic area with what appears to be exclusive responses totals more than 100%.

Specific

1. (p2) reference to “….. device issue and data collection” is out of context and unclear. No description of what devices/technology are being utilized by patients and included as a target by this study in the Introduction section.

2. (p2) what is meant by “….across the wider i healthcare team.”?

3. (p3) for survey content area, what is the difference between “device” and “technology”.

4. (p3) there is no clear description within the methods section of subject inclusion/exclusion, response inclusion/exclusion (such as the need for completed/partial responses), timeline (dates), geographic distribution

5. (p5) Qualitative data summaries using descriptions like “several participants…” and “some participants” might be confusing to many readers.

6. (p6, Table 1). Clarify total number of responses so that % can be calculated for each response. Yes/No answer can be consolidated into “Use of device to monitor vital signs (n, %). Its unclear if device type categories should be exclusive for the first 3 choices, yet total more than 100%. Unclear how branching logic impacted %, since those not using devices could not respond to type and advice. % Totals within categories (such as Advise on interpreting spirometry readings) with exclusive responses exceed 100%.

7. (p8) Statement beginning “Whilst we recommend….” Is confusing and should be reworded. What is meant by “CE marked”?

8. (p10) Denominators for the responses are not provided. Uncertain why ID numbers and genders are provided within the narrative.

9. (Figure 1) Contains overlapping data quantity labels within the figure so unable to interpret. No figure legend was provided

10. (Figure 2) Lacks quantitative data to determine how frequent the themes were presented. Perhaps this could be done by the size of the theme’s figure. A figure legend is needed to explain the connections.

11. (Figure 3) see above comments re: Figure 2

12. (Figure 4) not included in my copy

Reviewer #2: In the introduction, page 5, the authors indicate that digital devices should be used by ILD patients experiencing intractable breathlessness and psychological distress. The literature indicates that digital devices are recommended for all patients having ILD. It would be good if the authors clarify why they only recommend digital devices in critical conditions. Are digital devices useful in other conditions in ILD patients?

There are some typo mistakes that should be corrected.

The conclusion is very short for the work that the authors present. What are the conclusions about concerns or difficulties experienced by the patients? Any way to address those concerns and difficulties? Any suggestions about the periodicity to use the digital devices? Why?

The lower right part of figure 3 is practically illegible.

Reference 8 is from 2023, citation should be corrected.

Reviewer #3: This survey study aims to promote the use of digital devices for self-management among patients with interstitial lung disease (ILD). It provides valuable insights into the requirements for enhancing digital device use among ILD patients, a trend that has been growing recently. 

However, it is necessary to hypothesize why ILD patients need to self-manage using digital devices and the potential benefits their widespread use could bring to healthcare. ILD patients have traditionally been inclined towards self-monitoring and management, such as home oxygen therapy. The assumed benefits of monitoring disease status include early intervention, prevention of disease progression, and reduction in healthcare costs. The analysis and discussion could be improved, and ideally, the study should be expanded to include more subjects and reassess the following points:

1. Social factors significantly influence the use of digital devices. ILD patients are predominantly older men, often facing income and health literacy challenges. In this study, 33% of patients were under 60 and mostly women, which deviates from the typical ILD patient demographic. It is important to encourage self-management among those not included in this study, particularly those with low digital literacy and self-management difficulties.

2. The type of devices used likely varies based on the patient's disease stage and treatment. It is crucial to analyze the barriers to management and the necessary support, depending on the patient's background and disease stage in each study.

3. Despite the absence of economic-related questions in the questionnaire, the sudden mention of financial resources and cost-effectiveness issues in the author's eight points on page 18 is concerning. If this is a factor, the questionnaire and analysis should include patient income and perceptions of cost-effectiveness.

6. PLOS authors have the option to publish the peer review history of their article (what does this mean?). If published, this will include your full peer review and any attached files.

**Do you want your identity to be public for this peer review?** For information about this choice, including consent withdrawal, please see our Privacy Policy.

Reviewer #1: No

Reviewer #2: Yes: Cleva Villanueva

Reviewer #3: Yes: Yayoi Tetsuou TSUKADA

---

## [Decision Letter · Decision Letter 1]

13 Nov 2023

Supporting self-management for patients with Interstitial Lung Diseases: Utility and Acceptability of Digital Devices

PDIG-D-23-00257R1

Dear Dr Russell,

We are pleased to inform you that your manuscript 'Supporting self-management for patients with Interstitial Lung Diseases: Utility and Acceptability of Digital Devices' has been provisionally accepted for publication in PLOS Digital Health.

Best regards,

Haleh Ayatollahi

Section Editor

PLOS Digital Health

Reviewer Comments (if any, and for reference):

Reviewer's Responses to Questions

**Comments to the Author**

1. If the authors have adequately addressed your comments raised in a previous round of review and you feel that this manuscript is now acceptable for publication, you may indicate that here to bypass the “Comments to the Author” section, enter your conflict of interest statement in the “Confidential to Editor” section, and submit your "Accept" recommendation.

Reviewer #1: All comments have been addressed

Reviewer #2: All comments have been addressed

2. Does this manuscript meet PLOS Digital Health’s publication criteria? Is the manuscript technically sound, and do the data support the conclusions? The manuscript must describe methodologically and ethically rigorous research with conclusions that are appropriately drawn based on the data presented.

Reviewer #1: Yes

Reviewer #2: Yes

3. Has the statistical analysis been performed appropriately and rigorously?

Reviewer #1: Yes

Reviewer #2: N/A

4. Have the authors made all data underlying the findings in their manuscript fully available (please refer to the Data Availability Statement at the start of the manuscript PDF file)?

Reviewer #1: Yes

Reviewer #2: Yes

5. Is the manuscript presented in an intelligible fashion and written in standard English?

Reviewer #1: Yes

Reviewer #2: Yes

6. Review Comments to the Author

Reviewer #1: Addressed major comments adequately. Revision greatly improved.

Reviewer #2: The authors addressed all the questions and changed the manuscript accordingly. The manuscript follows all the requirements of Plos Digital Health and all data are available. Figure 3 is not readable (lower part in the right), it would be good if letters go in black. The manuscript is acceptable to be published at Plos Digital Health.

7. PLOS authors have the option to publish the peer review history of their article (what does this mean?). If published, this will include your full peer review and any attached files.

**Do you want your identity to be public for this peer review?** For information about this choice, including consent withdrawal, please see our Privacy Policy.

Reviewer #1: No

Reviewer #2: **Yes: **Cleva Villanueva
